# Comparative multi-tissue profiling reveals extensive tissue-specificity in transcriptome reprogramming during thermal adaptation

**Noushin Hadadi[1,2†], Martina Spiljar[1,2†‡], Karin Steinbach[3], Melis Çolakoğlu[1,2], Claire Chevalier[1,2], Gabriela Salinas[4], Doron Merkler[3], Mirko Trajkovski[1,2]***

[1]Department of Cell Physiology and Metabolism, Faculty of Medicine, Centre Medical Universitaire (CMU), University of Geneva, Geneva, Switzerland; [2]Diabetes center, Faculty of Medicine, University of Geneva, Geneva, Switzerland; [3]Department of Pathology and Immunology, Faculty of Medicine, Centre Medical Universitaire (CMU), University of Geneva, Geneva, Switzerland; [4]NGS- Integrative Genomics Core Unit (NIG), Institute of Human Genetics, University Medical Center, Göttingen, Germany

**\*For correspondence:**
mirko.trajkovski@unige.ch

[†]These authors contributed equally to this work

**Present address:** [‡]Evergrande Center for Immunologic Diseases and Ann Romney Center for Neurologic Diseases, Harvard Medical School and Brigham and Women's Hospital, Boston, United States

**Competing interest:** The authors declare that no competing interests exist.

**Abstract** Thermal adaptation is an extensively used intervention for enhancing or suppressing thermogenic and mitochondrial activity in adipose tissues. As such, it has been suggested as a potential lifestyle intervention for body weight maintenance. While the metabolic consequences of thermal acclimation are not limited to the adipose tissues, the impact on the rest of the tissues in context of their gene expression profile remains unclear. Here, we provide a systematic characterization of the effects in a comparative multi-tissue RNA sequencing approach following exposure of mice to 10 °C, 22 °C, or 34 °C in a panel of organs consisting of spleen, bone marrow, spinal cord, brain, hypothalamus, ileum, liver, quadriceps, subcutaneous-, visceral- and brown adipose tissues. We highlight that transcriptional responses to temperature alterations exhibit a high degree of tissue-specificity both at the gene level and at GO enrichment gene sets, and show that the tissue-specificity is not directed by the distinct basic gene expression pattern exhibited by the various organs. Our study places the adaptation of individual tissues to different temperatures in a whole-organism framework and provides integrative transcriptional analysis necessary for understanding the temperature-mediated biological programming.

## Editor's evaluation

This study gene profiled multiple tissues from mice housed at room temperature, cold and mildly warm. The availability of the gene expression datasets should be of interest to the field and may open up new research avenues regarding temperature responses in different tissues.

## Introduction

Over the last decade and since the discovery of active brown adipose tissue (BAT) presence in humans, cold acclimation for stimulating thermogenesis has gained great interest as an intervention leading to increased energy expenditure (*Chondronikola et al., 2016*; *Nedergaard et al., 2007*). The association between activating the thermogenic program of BAT and metabolic health has been extensively studied (*Waldén et al., 2012*). Most of these studies are conducted on mouse models and different cold exposure interventions have been used to alter adipose tissue activity and metabolism (*Peres*

**eLife digest** Humans, mice and most other mammals are constantly exposed to fluctuations in the temperature of their environment. These fluctuations cause striking metabolic effects in the body, for example, exposure to cold promotes burning of calories to generate heat, thereby reducing how much fat accumulates in the body. On the other hand, warmer temperatures strengthen the bones and protect against a bone disease known as osteoporosis. As such, it has been suggested that exposure to alternating warm or cold temperatures could be a potential lifestyle intervention that conveys various benefits to our health.

Our body stores fat in tissues known as adipose tissues, which are found all over the body including under the skin and around our major organs and muscles. Exposure to cold triggers changes in the activities of some genes in the adipose tissues to burn more calories. But it remains unclear how temperature affects the activities of other organs with respect to their expression of genes in the whole-body context.

Hadadi, Spiljar et al. used an RNA sequencing approach to study the activities of genes in various tissues of mice exposed to cold (10°C), room temperature (22°C), or mild warm (34°C). The experiments revealed numerous genes whose levels were different in the various organs and temperatures tested.

Overall, adipose tissues experienced the biggest changes in gene levels between different temperatures, followed by tissues involved in immune responses, and the brain and spinal cord tissues. Each organ changed gene expression levels in its own way. , and this was not due to the different intimate gene expression profile between the various organs.

These findings improve our understanding of how changes in temperature affect mammals by putting the responses of individual tissues into the context of the whole body. Hadadi, Spiljar et al. also generated a web-based, free-to-use application to allow others to view and further analyze the data collected in this work for gene levels in the various organs of interest.

*Valgas da Silva et al., 2019*). The BAT is present at distinct anatomical sites, including the interscapular (iBAT), perineal and axillary depots. The white adipose tissue (WAT) stores energy in form of triglycerides, and is found throughout the body. Its largest compartments are the subcutaneous and visceral adipose tissues (SAT and VAT, respectively). Following prolonged cold, brown fat-like cells also emerge in SAT (known as 'beige' cells) in a process referred to as white fat browning. Brown and beige fat-associated thermogenesis account for 2–17% of the total 12–20% of energy that is expanded daily (*Tan et al., 2011*), indicating that other tissues may also contribute (*van Marken Lichtenbelt and Schrauwen, 2011*), as has recently been shown for the liver (*Abumrad, 2017*; *Simcox et al., 2017*) and muscle (*Rowland et al., 2015*). However, the contribution of other organs in the overall adaptation of the organism to cold exposure is less understood (*Omran and Christian, 2020*).

The hypothalamus is the central regulating unit in the brain for maintenance of energy homeostasis, including body temperature. During cold, sympathetic signals are sensed by adipocytes via their beta-adrenergic receptors, which results in BAT activation (*Cannon and Nedergaard, 2004*; *Chechi et al., 2013*; *Stojanović et al., 2018*). Immune cells are also implicated in the effects of cold exposure in the adipose tissues (*Hotamisligil, 2017*; *Kohlgruber et al., 2016*; *Molofsky et al., 2013*; *Omran and Christian, 2020*; *Qiu et al., 2014*; *Stojanović et al., 2018*), which harbor an antiinflammatory immune profile under cold. In obesity, the fat predominantly contains inflammatory immune cells linked to systemic low-grade inflammation (*Hotamisligil, 2017*; *Kohlgruber et al., 2016*). How the systemic immune state at cold or warm temperature, however, remains largely elusive. While immune cells reside in a variety of organs, the greatest immune hubs are the primary lymphoid tissues where immune cells are generated (e.g. bone marrow), and the secondary lymphoid tissues where they are activated and expanded (e.g. spleen). Small stretches of lymphoid tissues also reside within other organs, including the intestine that contains the gut-associated lymphoid tissues (GALT). We (*Chevalier et al., 2020*; *Chevalier et al., 2015*; *Spiljar et al., 2021*) and others (*Simcox et al., 2017*) observed that cold (4–10°C), and warm exposures close to thermoneutrality (33–34°C) also affect organs apart from the adipose tissues, including the intestine, bone, and immunologic tissues. The ambient temperatures where metabolic rate is at a minimum is the usual definition of a thermoneutral

zone, and it is in the interval of 29 °C in the light phase and up to 33 °C during dark (**Škop et al., 2020**). The typical housing of mice is at room temperature (RT) of 22 °C, which is below their thermoneutrality, and a substantial amount of their energy expenditure is devoted to maintaining core body temperature. The above data suggest that thermal variations exert a whole-body functional reprogramming; however, no systematic transcriptomics analysis has addressed to what extent organs undergo changes induced by temperatures below the thermoneutral zone. It is also not clear whether various organs display conserved, or tissue-specific transcriptional signatures.

RNA sequencing (RNASeq) is the most widely used quantitative approach to assess the global gene expression and its alterations under different conditions, since it determines subtle molecular changes that may contribute to acquisition of certain phenotypes (**Carninci et al., 2005**; **Grada and Weinbrecht, 2013**). It has been shown that RNAseq can reflect tissue specificity; as such, highly regulated genes might represent characteristic functions of the tissues (**Breschi et al., 2016**; **Sonawane et al., 2017**). However, due to a deluge of data, particularly when several transcriptomic data sets are obtained, the combined analysis of several datasets (meta-analysis) remains challenging (**Sudmant et al., 2015**). This analysis becomes even more challenging if one aims to interconnect the variations in the meta-data with the complex molecular basis of phenotypic changes.

In this study, we conducted a systematic transcriptomics analysis across 11 mouse tissues at RT and cold (10 °C), and across additional 7 tissues at mild warm close to the mouse thermoneutral zone (33–34°C). We establish a common expression signature of differentially expressed genes in BAT during various cold exposure experimental setups across seven studies (six previously published RNAseq datasets and this work). Further, we provide a comprehensive resource dataset of comparative transcriptomics across 11 mouse tissues describing the transcriptional landscape at room temperature (RT) and at 10 °C, and supplement these analyses with a multi-tissue transcriptomics profiling at 34 °C. We systematically investigate how differential expression impacts specific cellular functions across the tissues and identify shared, specific, and inversely regulated gene signatures and gene set enrichments under cold exposure and at 34 °C compared to RT housing. Our work shows that adipose tissues undergo most severe transcriptome changes, followed by the immune tissues and the CNS cluster. With this resource and the applied bioinformatics methods, we characterize tissue-specific expression patterns and detect temperature-dependent gene expression profiles; provide insights

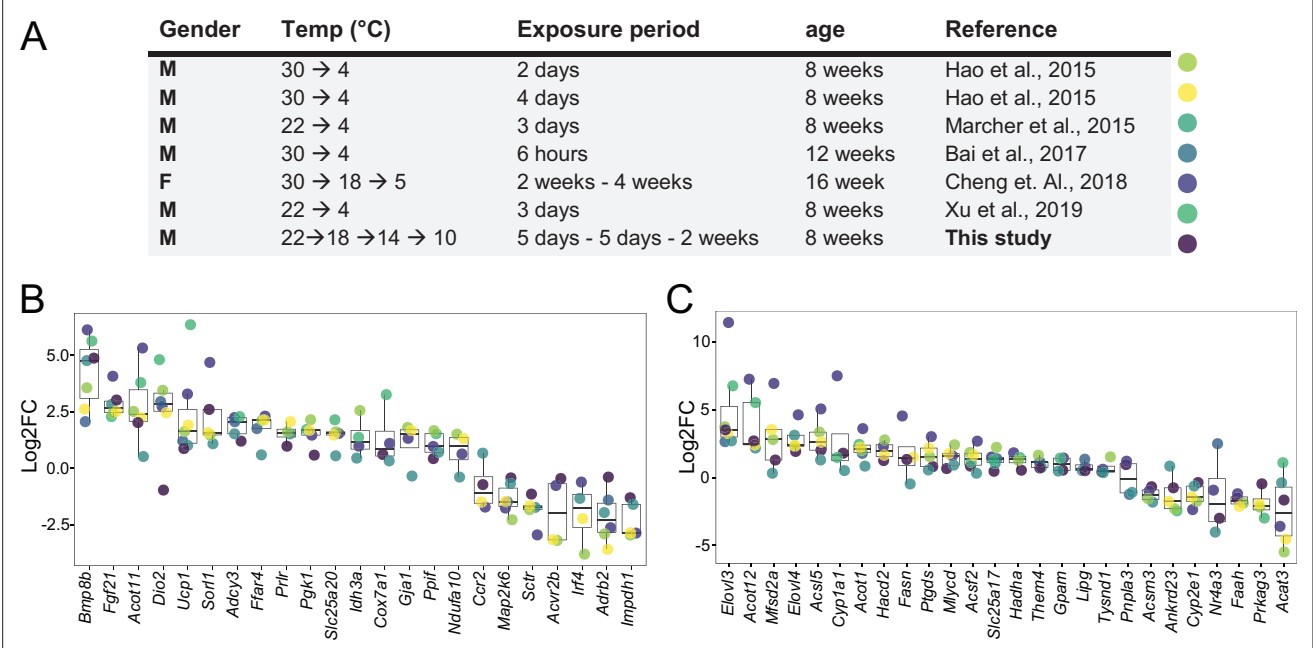

**Figure 1.** Log2FC of brown adipose tissue (BAT) biomarkers across seven datasets. (**A**) Table of seven cold exposure studies with their corresponding experimental conditions. (**B**) Log2FC of 23 selected BAT thermogenic markers. (**C**) Log2FC of 26 selected fatty acid metabolism gene markers. The markers in (**B**) and (**C**) were selected as being significantly (p<0.05) regulated in at least four datasets. Each dot corresponds to the log2FC of the indicated gene and colors specify the corresponding study as designated in (**A**).

into the tissue-specific adaptation mechanisms associated with temperature variations; and place the adaptive role of each tissue in a whole-organism perspective to comprehend the tissue-specific organization of the biological processes after 12 °C increase or decrease of environmental temperature compared to RT housing.

## Results

### Effectiveness of 10°C for activation and recruitment of the thermogenic capacity of brown adipose tissue (BAT)

We first investigated the effectiveness of a mild cold intervention using 10 °C on the interscapular brown adipose tissue (iBAT) following a gradual decrease of the temperature, to mimic the environmental temperature decrease that typically happens in nature. We combined the iBAT thermogenic biomarkers introduced in the literature (*Perdikari et al., 2018*) with those genes in the Gene Ontology (GO) database which are annotated to any GO term related to adaptive thermogenesis. This resulted in a list of 148 potential thermogenesis-induced genes, of which 120 genes were significantly differentially expressed in at least one of the seven studies *Supplementary file 1*. We next compared the regulation of these genes in our study to the six other publicly available transcriptomic datasets, where C57BL/6J male mice were used in a cold exposure intervention and compared to littermates housed either at RT or at 30 °C (*Bai et al., 2017*; *Cheng et al., 2018*; *Hao et al., 2015*; *Marcher et al., 2015*; *Shore et al., 2013*; *Xu et al., 2019*; *Figure 1a*). Several parameters such as gender, initial temperature, the cold temperature, the duration of the cold exposure, and the nature of the exposure (acute or chronic) are varying across these studies.

Within the identified genes across the seven different studies, we found *Ucp1* and *Bmp8b* being increased in all seven datasets, followed by *Dio2* and *Fgf21* that were regulated in six studies (*Supplementary file 1*). The cold exposure study of *Cheng et al., 2018* covered a maximum of 75 of the total 120 differentially expressed genes from all the seven studies, followed by *Marcher et al., 2015* with 63 and this study with 55, while the study by *Bai et al., 2017* identified the least with 13. Out of the 120 genes, we selected 23 genes that were significantly regulated in at least four datasets and compared their log2FC across the seven studies (*Figure 1B*). Despite the notable differences in the experimental conditions, both the number of thermogenesis-induced genes and the magnitude of their change in iBAT at 10 °C were in agreement with those obtained at colder temperatures.

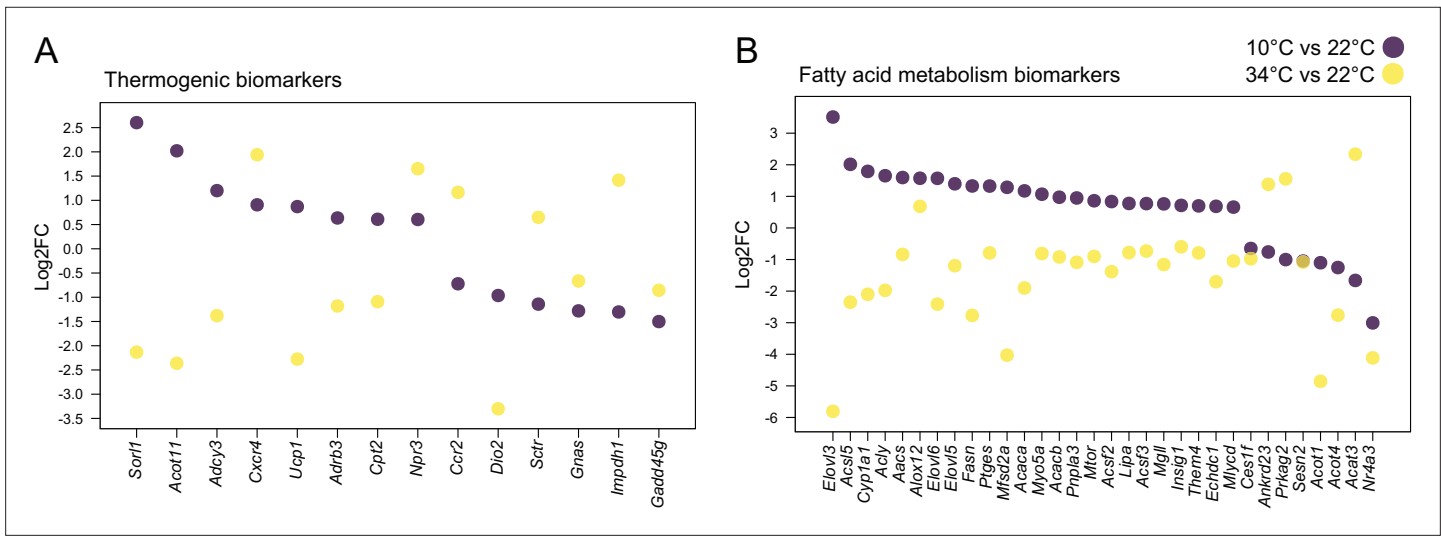

**Figure 2.** Cold and mild warm exposures induce opposite gene expression changes in BAT. Thermogenic (**A**) and fatty acid metabolism (**B**) biomarkers in BAT of mice exposed to 10 °C vs. 22 °C, and 34 °C vs. 22 °C controls. All shown genes are significantly changed (p<0.05) and their Log2FC was calculated based on replicates with p-values listed in *Supplementary file 6*.

The online version of this article includes the following figure supplement(s) for figure 2:

**Figure supplement 1.** Heatmap of the thermogenic.(**A**) and fatty acid metabolism (**B**) biomarkers in BAT and SAT of mice as in *Figure 2* showing the individual values.

We further focused on 292 genes involved in fatty acid metabolism (as annotated in GO database), a process of critical importance for the cold-induced thermogenesis *Supplementary file 2*. We identified 190 genes which were significantly differentially expressed in at least one of the seven studies. Only Elovl3 was significantly changed in all seven studies, and we observed a similar trend for the number of fatty acid metabolism-related genes across seven studies, with the maximum number of 127 differentially expressed genes in *Cheng et al., 2018*, 105 in this study, 101 in the study by *Marcher et al., 2015* and the lowest number with 10 in *Bai et al., 2017*. Comparing the log2FC of the 26 genes that were significantly changed in at least four datasets (*Figure 1C*) shows that consistent with the thermogenesis-regulated genes (*Figure 1B*), the regulation of fatty acid metabolism related-genes in our study using 10 °C is in agreement with the other studies. These comparisons (both the number of differentially expressed genes and the magnitude of their changes) suggest that 10 °C used in this study induces gene expression alterations that are similar to the harsher cold exposures.

To better understand BAT physiological changes which are associated with thermal adaptation, we further investigated gene expression changes after 4 weeks exposure to 34 °C compared to 22 °C. 2,126 genes were downregulated and 1118 were upregulated (p<0.05 and |log2FC|>0.6) in BAT from mice kept at 34 °C compared to 22 °C-kept littermates *Supplementary file 3*. Notably, these changes were more pronounced compared to the 767 upregulated genes and 988 downregulated genes (p<0.05 and |log2FC|>0.6) *Supplementary file 4* when comparing 10–22°C, although in both thermal adaptations the temperature change is 12 °C. 76% of the 606 commonly changed genes between 34 °C and 10 °C were regulated in the opposite direction (*Supplementary files 5 and 6*). A total of 14 thermogenic and 32 fatty acid metabolism biomarkers were found among the 606 genes, with 82% of fatty acid biomarkers and 65% of thermogenic biomarkers showing reverse regulation patterns consistent with the direction of the temperature change (*Figure 2*, *Figure 2—figure supplement 1*, *Supplementary files 5 and 6*). Examples involve *Ucp1* with log2FC of 0.87 and –2.27, *Sorl1* with log2FC of 2.61 and –2.13, *Elovl3* with log2FC of 3.51 and –5.81, *Elovl6* with log2FC of 1.57 and –2.41, and *Acot11* with log2FC of 2.02 and –2.36 when comparing 22–10°C, and 34–22 °C, respectively. These results suggest that the regulation of the gene expression is largely proportional to the temperature gradient, with more pronounced changes at higher housing temperatures.

## A comprehensive mouse multi-tissue transcriptomic resource at 10°C, 22°C, and 34°C

To identify how thermal adaptation affects the tissue-specific signatures and whether it induces common gene expression changes across various tissues, we performed RNAseq on interscapular BAT (iBAT), bone marrow, brain, hypothalamus, ileum, liver, inguinal SAT (ingSAT), spinal cord, spleen, quadriceps and perigonadal (epididymal) VAT (pgVAT) from mice exposed to 10 °C for 2 weeks after an acclimatization at 18 °C and 14 °C for 5 days each, and compared them to room temperature-kept controls (*Figure 3A*). We complemented these analyses with the changes in gene expression of ingSAT, pgVAT, iBAT, brain, ileum, liver and quadriceps from mice housed at 34 °C for 4 weeks and compared them to the RT-kept controls.

All samples were subjected to quality control and clustering analysis, and we restricted our analyses to all genes with counts per million greater than zero. We performed two principal component analysis (PCA), using a combined dataset with 11,224 genes that were identified across all tissues once with the 11 samples at 22 °C and 10 °C and in a second PCA, with 7 tissue samples at 22 °C and 34 °C (*Figure 3B and D*). The first PCA revealed two strong clusters of physiologically related tissues (*Figure 3B*): The adipose tissue cluster with iBAT, ingSAT, and pgVAT, and the CNS cluster with hypothalamus, brain, and spinal cord samples. The bone marrow samples clustered relatively close to the spleen samples, which can be considered as an immune tissue cluster. The liver and quadriceps samples gathered close to the adipose tissues, while the ileum samples clustered between adipose tissue and spleen samples. Generally, the samples clustered rather by tissue and not by treatment (22 °C vs. 10 °C or 34 °C). We observed similar results in the second PCA where the iBAT, ingSAT, pgVAT, and quadriceps samples clustered together.

To understand the propensity of samples from 10 °C or 34 °C to RT control mice per given tissue, we performed PCA on each tissue separately (*Figure 3C and E*), and observed a different degree of clustering. Specifically, we detected a marked separation of RT and 10 °C samples in iBAT, opposed to a milder separation in hypothalamus samples, implying that the cold exposure has varying effects

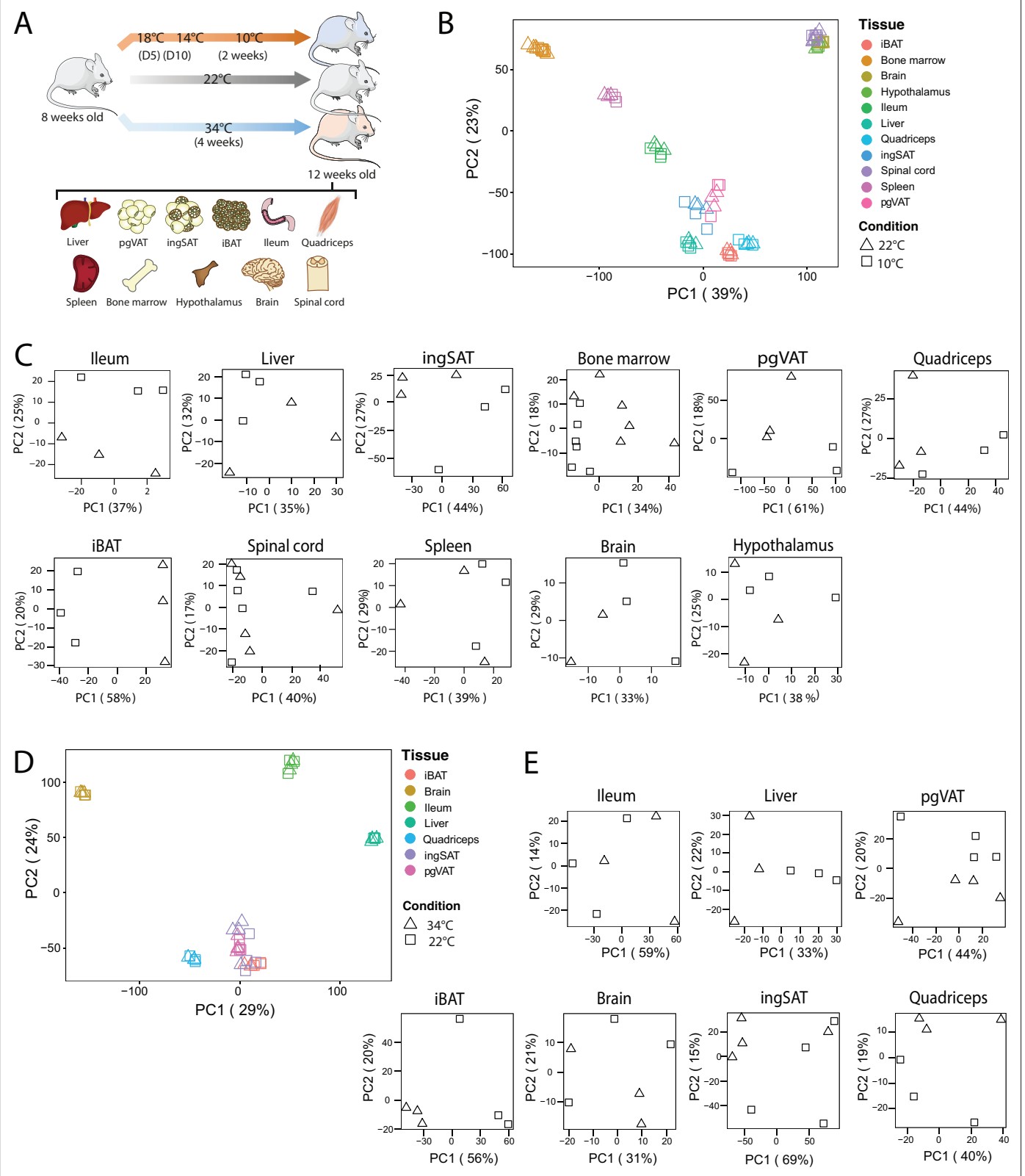

**Figure 3.** QC and clustering of the biological samples and tissues at 10 °C, 22 °C, and 34 °C. (**A**) Experimental setup: 8-week-old C57BL/6 J mice were exposed to 34 °C for 4 weeks, 10 °C for 2 weeks following acclimatization at 18 °C, and 14 °C for 5 days each, or kept at room RT (22 °C) during the exposure of the respective temperature-challenged group, and their tissues were harvested for RNA sequencing. (**B**) Principal component analysis (PCA) of the transcriptome across the 11 tissues at RT and 10 °C of mice as in (**A**). Physiologically close tissues are distinguished by global gene expression

*Figure 3 continued on next page*

Figure 3 continued

patterns. For 10 °C samples, each dot represents a pool of two mice, except for the spinal cord and bone marrow, where each dot represents one mouse. (**C**) PCA of samples for each tissue from mice as in (**B**). (**D**) Principal component analysis (PCA) of the transcriptome across the 7 tissues of mice as in (**A**) exposed to 34 °C or RT for 4 weeks. (**E**) PCA of samples for each tissue from mice as in (**A**).

depending on the tissue. Similarly, 34 °C exposure caused the most pronounced segregation in iBAT compared to RT controls, indicating that iBAT displays the highest degree of temperature-dependent alterations among the analyzed tissues.

Additionally, biological replicates across all tissues and the relationship between the tissues from 10 °C-kept mice and their RT controls were analyzed using hierarchical clustering with pairwise Pearson correlation. We observed high Pearson correlation (0.92–1) between samples from the same tissue but different mice, suggesting that inter-individual variation has little impact on the transcriptomic profiles and demonstrating high reproducibility of the data (*Figure 2—figure supplement 1*). Larger clusters, as for instance the one between ingSAT, iBAT, and pgVAT, both at RT and 10 °C, indicate that physiologically close tissues show high similarity in terms of their global gene expression profiles.

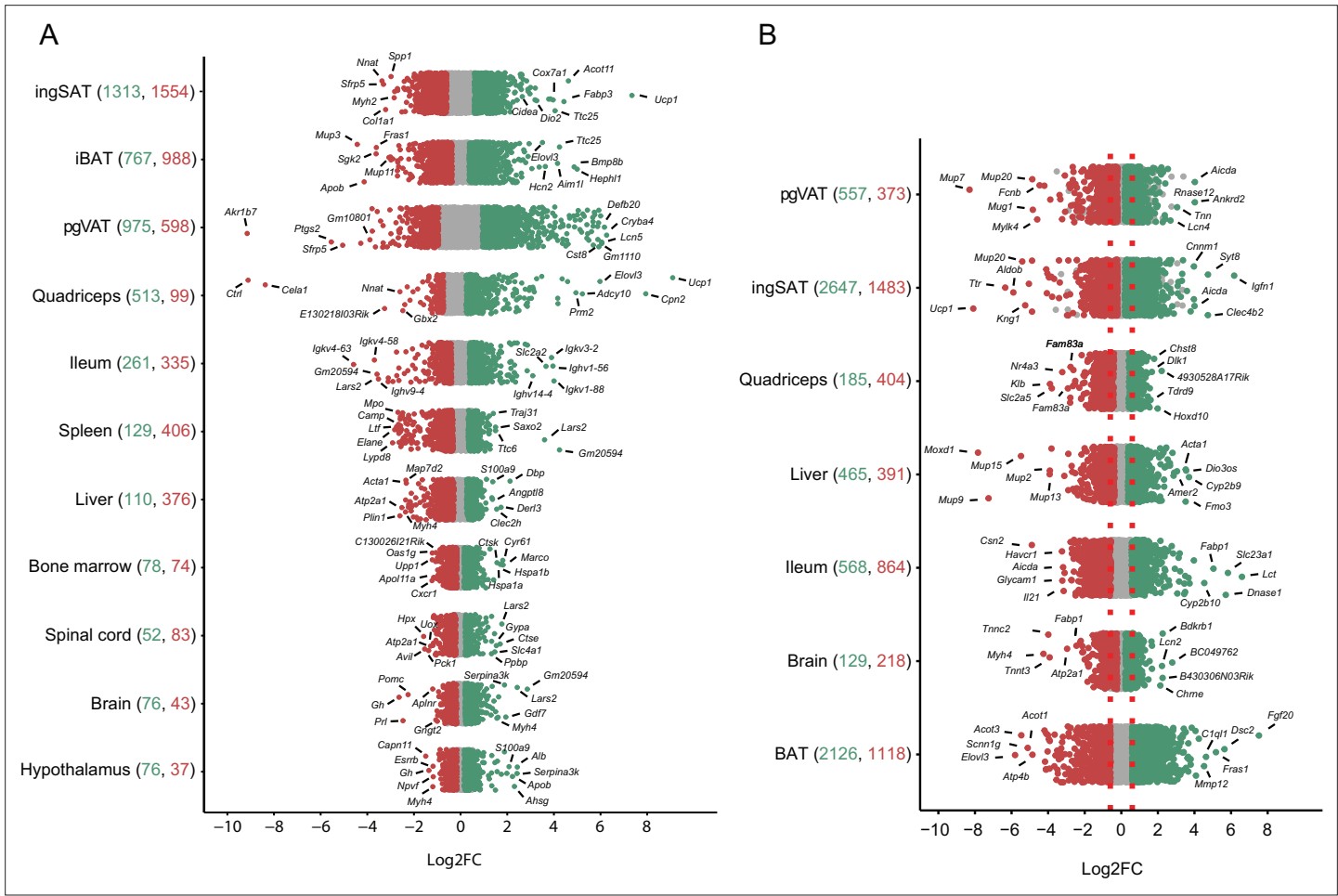

**Figure 4.** Distribution and degree of differential gene expression when comparing 10 °C to 22 °C and 34 °C to 22 °C. Log2FC of all genes across the tissues of mice exposed to 10 °C vs. 22 °C (**A**) and 34 °C vs. 22 °C (**B**). Red and green colored dots indicate p<0.05, whereas grey dots indicate p>0.05. Numbers in parentheses on the x-axis indicate the number of significantly up- (green) and down- (red) regulated genes. Gene names are shown for the 5–7 most up- and down-regulated genes with highest log2FC.

## Differentially expressed genes and enriched GO biological terms in tissues of mice exposed to 10°C and 34°C compared to 22°C controls

To further characterize the transcriptional thermal adaptation, we first assessed the number of differentially expressed genes and the magnitude of their changed expression across different tissues (*Figure 4*, *Supplementary file 4*), followed by GO gene set enrichment analysis of the identified differentially expressed genes. In the 10 °C exposure scenario, in all 11 tissues collectively we found 4350 genes as up- and 4593 genes as down-regulated (p-value <0.05, |FC|>1.5), while 785 genes were inversely regulated, that is, increased in a given tissue(s) and decreased in others. This inverse regulation may indicate that the regulation of certain genes occurs in a tissue specific manner. Physiologically close tissues that clustered together (*Figure 3B*) shared similar numbers of differentially expressed genes. IngSAT underwent most transcriptomic changes upon exposure to cold with 2867 significantly differentially expressed genes, followed by iBAT and pgVAT, while brain and hypothalamus showed the least transcriptomic changes with 119 and 113 total differentially expressed genes, respectively (*Figure 4A*). These results suggest that although adipose tissues are the most affected by cold exposure, all organs undergo changes that are reflected in their tissue-specific transcriptomic profiles.

Moreover, the magnitude of gene expression changes (log2FC) showed an analogous ordering depending on the tissue. Adipose tissues had the highest degree of differential expression in both upregulated (e.g. log2FC of *Ucp1* in ingSAT 7.34, in muscle 8.82 or *Elovl3* in muscle 7.73) and down-regulated genes (e.g. log2FC of *Akr1b7* in pgVAT –9), while spinal cord, brain, and hypothalamus scored the mildest changes (*Figure 4*, *Supplementary file 4*).

In the 34 °C exposure scenario, the most severe expression changes occured in BAT with 2126 upregulated and 1118 downregulated genes (*Figure 4B*, *Supplementary file 3*). Comparing the differentially expressed genes between 10 °C and 34 °C exposures across the 7 common tissues, we found a high number of commonly regulated genes per same tissue. The highest number of commonly regulated genes was found in SAT where 854 genes were regulated both at the 10 °C and 34 °C exposures, amongst which 66% were altered in opposite directions (*Supplementary file 5*). Interestingly, similar to the observation in BAT (*Figure 2A*), the magnitude of the regulation was bigger in 63% of the regulated genes in the same direction (285 genes). A total of 41 thermogenic and 38 fatty acid metabolism biomarkers were found in the 854 gene list and more than 90% of these genes were reversely regulated (*Figure 5* and *Supplementary file 1*).

The BAT scores the second in the number of commonly regulated genes (details in the BAT section), followed by VAT with 193 genes (62% regulated in the opposite directions), ileum with 172 genes (18% regulated in the opposite direction), quadriceps with 120 genes (92% regulated in opposite

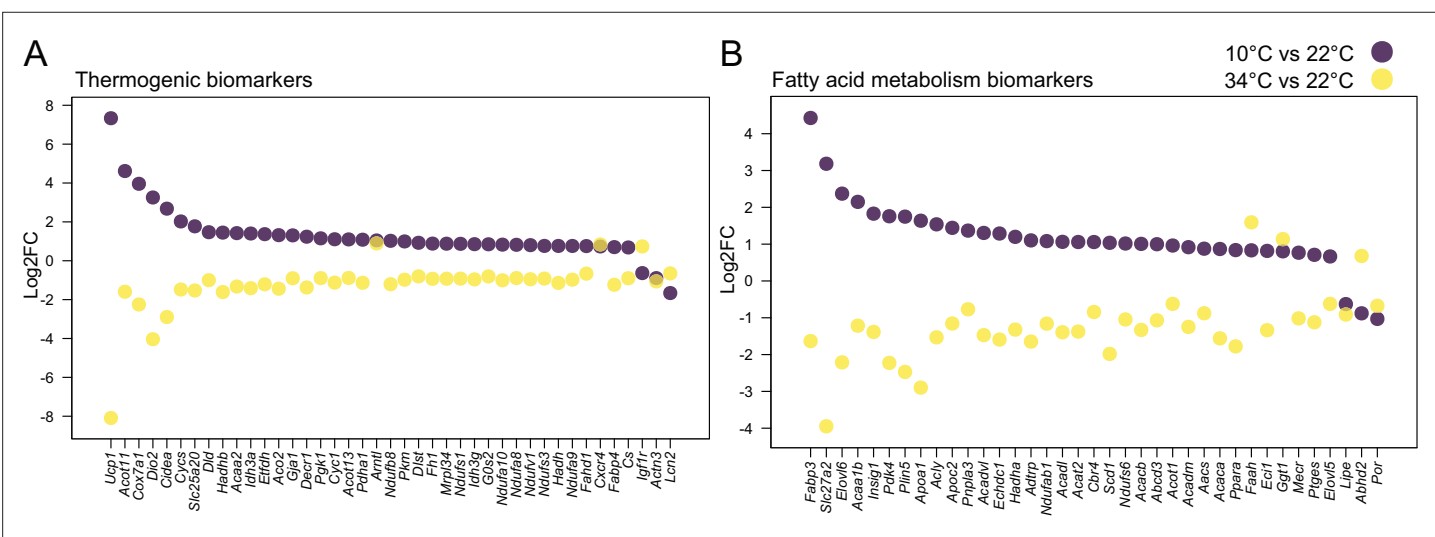

**Figure 5.** Thermogenesis and fatty acid metabolism are altered in opposing directions during cold and warm. Thermogenic (left) and fatty acid metabolism (right) biomarkers in SAT when comparing 10°C to 22°C and 34°C to 22°C. All shown genes are significantly changed (p<0.05) and their Log2FC was calculated based on replicates with p-values listed in *Supplementary file 6*.

**Table 1.** GO-based gene set enrichment analysis when comparing 10 °C vs 22 °C and 34 °C vs 22 °C.

| | Significantly enriched GO terms (p-value <0.05) | | | |
|---|---|---|---|---|
| Tissue | # of upregulated GO terms at 10 °C | # of downregulated GO terms at 10 °C | # of upregulated GO terms at 34 °C | # of downregulated GO terms at 34 °C |
| ingSAT | 83 | 178 | 264 | 177 |
| Quadriceps | 117 | 23 | 14 | 75 |
| pgVAT | 42 | 73 | 93 | 54 |
| iBAT | 43 | 53 | 175 | 82 |
| Spleen | 1 | 55 | - | - |
| Hypothalamus | 30 | 14 | - | - |
| Ileum | 25 | 12 | 59 | 187 |
| Liver | 3 | 27 | 64 | 19 |
| Bone marrow | 18 | 6 | - | - |
| Brain | 6 | 11 | 12 | 36 |
| Spinal cord | 8 | 5 | - | - |

direction), liver with 66 genes (25% in opposite direction), and brain with 6 genes (0% regulated in opposite direction) (*Supplementary file 5*).

These results suggest that the regulation of the adipose tissue and quadriceps, which were closely clustered together, share a similarly high degree of regulation in opposite directions when 10 °C and 34 °C are compared. This observation is contrary to the liver, ileum, and brain (more internal organs) where the responses to 10 °C and 34 °C are quite similar and the gene regulations are mostly in the same direction. This could be explained by the fact that the thermogenic and fatty acid metabolism biomarkers which consistently showed opposite regulation patterns are mostly regulated in adipose tissues and quadriceps.

We next performed GO gene set enrichment analysis on the differentially expressed genes at 10 °C and at 34 °C for all tissues (|FC|>1.5 and p -value<0.05). Enriched gene sets with p-value <0.01 were chosen for further analysis, collectively consisting of 457 downregulated and 376 upregulated gene sets at 10 °C (comprising 637 unique GO terms) and 630 downregulated and 681 upregulated gene sets (comprising 894 unique GO terms) at 34 °C (*Table 1*). The top 50 up- and down-regulated GO terms in both 10 °C and 34 °C exposures are visualized in *Figure 6* (full list is shown in *Supplementary file 7*). The selection was made based on the p value in response to the thermal change, including both tissue-shared and tissue-specific pathways. As some 10 °C and 34 °C responsive gene sets were shared between tissues, we next sought to unravel the biological GO terms that exhibit tissue-specific, or common regulation patterns across the tissues.

Upon 10 °C exposure, ingSAT showed the highest degree of enrichment (261 GO terms), followed by quadriceps, pgVAT, and iBAT. The least changed tissues were spinal cord and brain with 17 and 13 differentially expressed gene sets, respectively (*Table 1*). This ordering was similar to the degree of gene expression changes (shown in *Figure 3*), with an exception of the hypothalamus (which ranked 5th) when the GO terms were used as criteria. Similar to the 10 °C exposure, ingSAT ranked the highest following 34 °C exposure, where in total 441 unique GO terms were regulated. In general, changes at 34 °C were more pronounced compared to the 10°C-induced alterations.

## Tissue-specific and tissue-shared gene and GO term signatures in response to 10°C and 34°C exposure

We identified 248 'commonly' regulated GO terms between 10 °C and 34 °C exposures compared to RT, out of which 80 pathways were solely upregulated at 10 °C and downregulated at 34 °C (*Supplementary file 8*). Examples mostly involve the thermogenic and lipid-associated functions such as 'mitochondrial respiratory chain complex I assembly', 'fatty acid biosynthetic process', 'lipid storage',

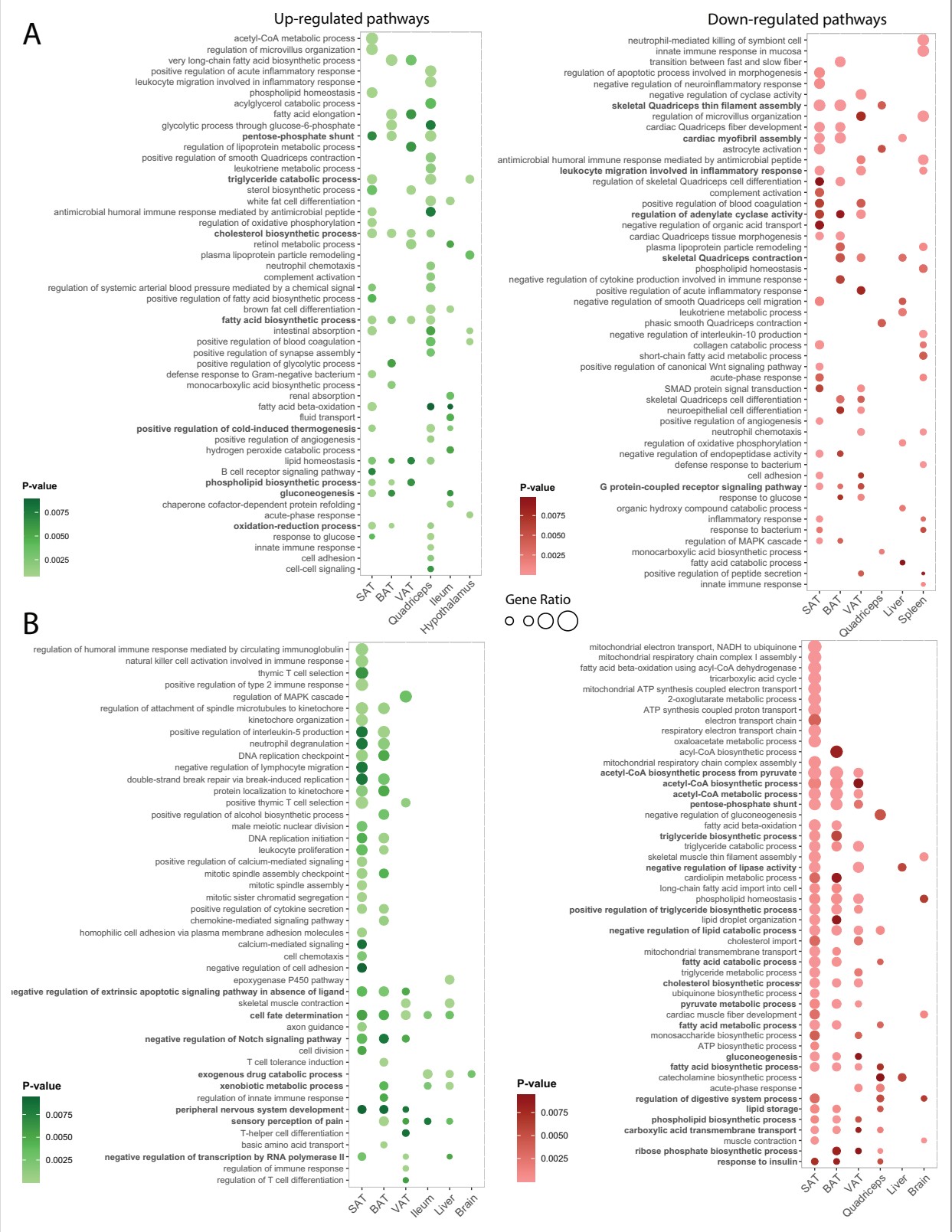

**Figure 6.** Visualization of the highly enriched gene sets indicates common enrichment within groups of tissues upon thermal adaptation. (**A, B**) Visualized are the top ranked 50 up- and down-regulated gene sets at 10 °C (**A**) and 34 °C (**B**) exposure. The p-value and the gene ratio (the ratio of differentially expressed genes to annotated genes in each gene set) were used as ranking criteria and the p-value weight is indicated with a gradient. The gene ratios are indicated by size of the dots and the exact values are provided in the supplementary *Supplementary file 7*. Gene sets that are commonly regulated in more than two tissues are highlighted in bold.

'gluconeogenesis', 'fatty acid beta-oxidation', and 'lipid droplet organization'.Forty-eight pathways showed opposite behavior being lowered at 10 °C and upregulated at 34 °C such as 'regulation of ion transmembrane transport' and 'fat-soluble vitamin metabolic process'. Twenty-six pathways were mutually downregulated at both 34 °C and 10 °C exposures, linked to 'muscle contraction', 'lipoprotein metabolic process', 'collagen fibril organization', and 'protein localization to extracellular region'. Twenty-two pathways were mutually increased at both conditions and included 'regulation of sensory perception of pain', 'positive regulation of cell population proliferation', and 'cell surface receptor signaling pathway'. The remaining 72 pathways displayed both down- and up-regulation profiles across different tissues and at each condition (*Supplementary file 8*). Interestingly, in this group were immune system related pathways such as 'immune response', 'adaptive immune response', 'regulation of T-helper 17 type immune response', etc (full list in *Supplementary file 8*) that were mostly enhanced in adipose tissues and quadriceps in both 10 °C and 34 °C exposures and were reduced in ileum and brain at 34 °C, and in spleen and ileum following 10 °C.

Out of the 637 altered GO terms, 502 (78%) were identified as tissue-specific regulated pathways (*Supplementary file 7*) at 10 °C, suggesting that the transcriptional cold response at the level of functional gene sets is to a large degree tissue-specific. The maximum number of tissues that share a given regulated gene set is four (*Figure 6A*). A closer inspection of the 135 commonly differentially expressed gene sets revealed that 38 were commonly downregulated between three (6 gene sets) and two (29 gene sets) tissues, while 4 gene sets were commonly upregulated between three (5 gene sets) and two (20 gene sets) tissues (*Figure 6A*, and the full list in *Supplementary file 7*). We observed a clear overrepresentation of the adipose tissue cluster and quadriceps in the shared differentially expressed gene sets, which suggests a common response in ingSAT, iBAT, pgVAT and quadriceps that extends to ileum, liver and spleen for certain GO terms. This indicates that the closely related tissues share a higher number of cold responsive gene sets.

We observed a slightly lower degree of tissue-specific regulated GO terms at the 34 °C exposure when compared to 10 °C, where out of the 894 overall regulated GO terms, 614 were tissue-specific (69%). The maximum number of tissues that shared a given regulated gene set was five, wherein 'cell fate determination' was commonly upregulated in ingSAT, pgVAT, iBAT, liver and ileum (*Figure 6B*). This was followed by sensory perception of pain that was commonly upregulated between four tissues. Six and 24 GO terms were commonly upregulated in three and two tissues, respectively. Similarly,

**Table 2.** Number of up- and down-regulated genes per tissue shown as tissue-specific up-regulation and down-regulation profiles at both 10 °C and 34 °C exposures.

The percentage of tissue-specific changes versus the total number of differentially expressed genes (as in *Figure 3*) is displayed in parenthesis.

|  | Tissue-specific upregulation profiles | | Tissue-specific downregulation profiles | |
|---|---|---|---|---|
|  | # of genes (% of tissue-specific) | | # of genes (% of tissue-specific) | |
| Tissue | 10 °C exposure | 34 °C exposure | 10 °C exposure | 34 °C exposure |
| ingSAT | 883 (67%) | 1621 (65%) | 995 (64%) | 911 (64%) |
| pgVAT | 719 (75%) | 139 (27%) | 570 (58%) | 144 (41%) |
| iBAT | 376 (49%) | 269 (74%) | 300 (74%) | 526 (54%) |
| Quadriceps | 292 (57%) | 32 (56%) | 258 (43%) | 75 (40%) |
| Ileum | 155 (59%) | 929 (57%) | 218 (65%) | 566 (88%) |
| Spleen | 91 (71%) | - | 212 (56%) | - |
| Liver | 77 (70%) | 142 (58%) | 56 (57%) | 154 (73%) |
| Bone marrow | 51 (65%) | - | 47 (64%) | - |
| Hypothalamus | 50 (66%) | - | 27 (63%) | - |
| Brain | 43 (57%) | 6 (67%) | 25 (68%) | 17 (68%) |
| Spinal cord | 27 (52%) | - | 22 (27%) | - |
| Total | 2764 (79%) | 3132 (59%) | 2730 (76%) | 2376 (62%) |

we identified 95 commonly downregulated GO terms upon 34 °C exposure in which 4 were shared between four tissues, 21 between three tissues and 14 between two tissues.

To gain further insights into the tissue-shared and tissue-specific enriched GO terms in response to 10 °C and 34 °C exposures, we classified the differentially expressed genes into two categories according to the number of tissues wherein they are differentially expressed (*Table 2*).

At the 10 °C exposure, altogether, 2764 genes were upregulated in only one tissue (79% of the total upregulated genes) and were defined as 'tissue-specific up-regulation profiles', while 2,730 genes were downregulated in a single tissue (76% of the total downregulated genes) and were identified as 'tissue-specific down-regulation profiles' (*Table 2*, *Supplementary file 4*). We observed widespread tissue-specific cold response in the adipose tissues where ingSAT ranked highest with 883 up- and 995 and downregulated genes. We further compared the tissue-specific differentially expressed genes with the total number of differentially expressed genes for each tissue, and indicated it as a ratio (*Supplementary file 2*). The highest percentage of tissue speci-ficity in terms of differentially expressed genes was seen in pgVAT with 74%, iBAT with 74%, and spleen with 71% tissue-specific upregulated genes. Considering the total number of differentially expressed genes (sum of up and downregulated genes), ingSAT, pgVAT and spleen showed the highest specificity with an average of 65 tissue-specific gene expression profiles (*Supplementary file 4*). Collectively, there was a comparable degree of tissue-specificity in upregulated versus downregulated profiles (79% compared to 76%). Similar to the GO terms results, in the 34 °C exposure experiment we observed a lower degree of tissue-specificity than in cold, with 59% of the upregulated, and 62% of downregulated genes in a tissue-specific manner (*Table 2*). Across temperatures, Ileum scored the highest with 88% tissue-specific downregulated gene expression profile (at 34 °C), followed by iBAT with 74% (at 10 °C), and liver with 73% of tissue-specific down-regulated genes (at 34 °C). In contrast to the 10 °C exposure, 34 °C caused a slightly higher degree of tissue-specificity in downregulated (62%) versus upregulated genes (59%) (*Supplementary file 2*). Notably, the downregulated genes in 34 °C exposure showed extensive overlap with the upreg-ulated genes at the 10 °C exposure (mostly exhibited in adipose tissues and quadriceps) (*Figures 2 and 5*, and *Supplementary file 5*).

We next focused on the genes with shared differential gene expression profiles within several tissues both at 10 °C and 34 °C. Interestingly, there was no overall common gene regulation set between all tissues at neither 10 °C, nor at 34 °C exposure. In the 10 °C exposure, the maximum tissue-shared gene signature was *Atp2a1*, which was decreased in 7 tissues (all adipose tissues, liver, bone marrow, hypothalamus, and spinal cord). The next top tissue-shared gene signatures were 4 genes that were commonly downregulated (*Nnat*, *Nr1d1*) and upregulated (*Hspa1b*, *Thrsp*) in 6 tissues. The rest of the tissue-shared genes were shared within groups of five, four, three, and two tissues (*Supplementary file 4*). In the 34 °C exposure, *Cyp26b1* was commonly upregulated in all tissues except brain, *Lonrf3* was upregulated in 5 tissues (pgVAT, liver, iBAT, ingSAT, and quadriceps) and *Zbtb16* was upregulated in pgVAT, ileum, iBAT, ingSAT, and liver. We further identified 8 genes which were commonly downregulated in 5 out of the 7 tissues, namely *Mrap*, *Orm1*, *Pnpla3*, *Scd1*, *Slc36a2* were downregulated in pgVAT, ileum, iBAT, ingSAT, and quadriceps; *Tlcd2* was downregu-lated in pgVAT, liver, iBAT, ingSAT, and quadriceps; and *Pfkfb3* in pgVAT, ileum, iBAT, ingSAT, and liver (*Supplementary file 4*).

The highest degree of tissue-shared upregulated genes was found within the adipose tissues, where 464 genes at 34 °C and 166 genes at 10 °C were upregulated in both ingSAT and iBAT. Next, 199 genes at 34 °C and 61 genes at 10 °C exposure were commonly upregulated in pgVAT and ingSAT, followed by 53 genes at 34 °C and 42 genes at 10 °C exposures being upregulated in both pgVAT and iBAT (*Supplementary files 3 and 4*). Likewise, ingSAT, iBAT, and pgVAT shared 69 genes at 34 °C and 32 genes at 10 °C that were commonly upregulated in the three tissues. Apart from the adipose tissues, the top-shared pairwise connections were related to the quadriceps, ileum, and liver. Similarly, the top-shared pairwise connections for downregulated genes were found in adipose tissues, where ingSAT and iBAT shared 263 downregulated genes at 34 °C, and 177 genes at 10 °C exposures. IngSAT and pgVAT had 117 shared downregulated genes at 34 °C and 71 genes at 10 °C exposure; followed by pgVAT and iBAT with 10 at 34 °C, and 33 at 10 °C exposure. Apart from the adipose tissues, quadriceps and ileum demonstrated highest pairwise connections.

## Shared-tissue regulated GO terms involve tissue-specific and tissue-shared genes

We next investigated whether the tissue-shared GO terms regulation is derived from the shared or the tissue-specific differentially expressed genes. We particularly focused on ingSAT, pgVAT, and iBAT, both at 10 °C and 34 °C. The Venn diagram of the first three most commonly up- and downregulated gene sets upon 10 °C exposure showed that they are largely derived by genes that are differentially expressed in a tissue-specific manner (*Figure 7A and B*, *Supplementary file 9*). This tissue specificity is more pronounced in the downregulated gene sets (*Figure 7B*). For example, in fatty acid biosynthesis process, only 4 genes are shared between the three tissues and the upregulation of the pathway mostly emerges from the tissue-specific genes in ingSAT (13 genes), pgVAT (9 genes) and iBAT (7 genes). This suggests that although different genes are upregulated in the three tissues, they might have a similar function that leads to a common functional response. Interestingly, the degree of tissue-specific gene upregulation is on average higher for pgVAT followed by iBAT and ingSAT. For the 34 °C exposure, we observed a similar trend of less tissue-shared genes and more tissue-specific genes which derived the regulation of tissue-shared pathways. Similar to the previous section, we observed a slightly lower tissue-specificity in response to 34 °C exposure compared to 10 °C exposure. This was more pronounced in the downregulated pathways at 34 °C exposure (lipid metabolism pathways).

## Tissue-specific response to cold is not orchestrated by the gene expression patterns

As the transcriptional response to 10 °C exposure caused higher tissue-specificity, we further scrutinized whether the strong tissue specificity in response to cold exposure is directed by tissue specificity in the overall gene expression patterns. We first accounted for the genes with minimum 5 raw counts as a filtering threshold, and observed that 19,881 genes are expressed at least in one tissue, wherein 10,853 genes (54% of expressed genes) were expressed in all 11 tissues (*Figure 8A*, *Supplementary file 10*). From these, 3507 genes (32%) were differentially expressed in at least one tissue (*Figure 8B*). Looking at these differentially expressed genes, irrespective of the consistency in the direction of change, we found that 2435 genes (over 70%) were altered in only one tissue, 755 (over 21%) in two, and 220 (over 6%) commonly differentially expressed in three tissues. The remaining 2.1% of differentially expressed genes were commonly changed in four or more tissues (*Figure 8C*). These data demonstrate high tissue-specificity in the transcriptional response to cold also among the commonly expressed genes. Moreover, 1508 genes (8% of expressed genes) were expressed in only one tissue (*Figure 8A*). Spinal cord with 375 genes showed the most tissue-specificity in term of expressed genes, followed by pgVAT (328 genes), bone marrow (160 genes), ileum (139 genes), spleen (137 genes), liver (93 genes), hypothalamus (78 genes), brain (49 genes), iBAT (7 genes), and ingSAT (9 genes). Interestingly, only 66 genes of the 1623 tissue-specific expressed genes were significantly differentially expressed upon cold exposure (*Figure 8B*). Together, this indicates that the high degree of tissue-specificity in the transcriptional responses during cold exposure is not orchestrated by the global expression pattern exhibited by the various organs.

To further challenge these conclusions, and given the importance of the thermogenic and fatty acid metabolism biomarkers in the cold response, we specifically analyzed and compared the tissue specificity on both global expression and regulation levels of the 148 thermogenic and 292 fatty acid metabolism biomarkers (*Supplementary files 1 and 2*) across the 11 tissues (*Figure 9*). All the 148 thermogenic markers are expressed (minimum threshold of 5 raw counts) at least in two tissues, that is, there is no gene that is specifically expressed only in one tissue. Surprisingly, we found only 32 genes which are not expressed in all 11 tissues, which is in agreement with the global expression analysis. Liver contributes the most to this list by not expressing 16 of these 32 genes (*Figure 9A*, *Supplementary file 11*). Although all the 148 thermogenic biomarkers are expressed across most of the 11 tissues, only 97 genes exhibit significant changes (p-value <0.05, |FC|>1.5) upon cold exposure (*Figure 9C*), wherein 71 belong to adipose tissues. ingSAT has the highest number of differentially expressed thermogenic biomarkers, followed by iBAT and pgVAT (65, 31, and 25 differentially expressed genes, respectively), while the brain showed no thermogenic biomarker changes (*Figure 9C*).

Out of the 292 fatty acid metabolism biomarkers, 278 genes are expressed (minimum threshold of 5 raw counts) in at least one tissue. Similarly, most of the expressed genes (75 %) were expressed in the 11 tissues and we did not observe a strong tissue specificity on the level of gene expression in fatty

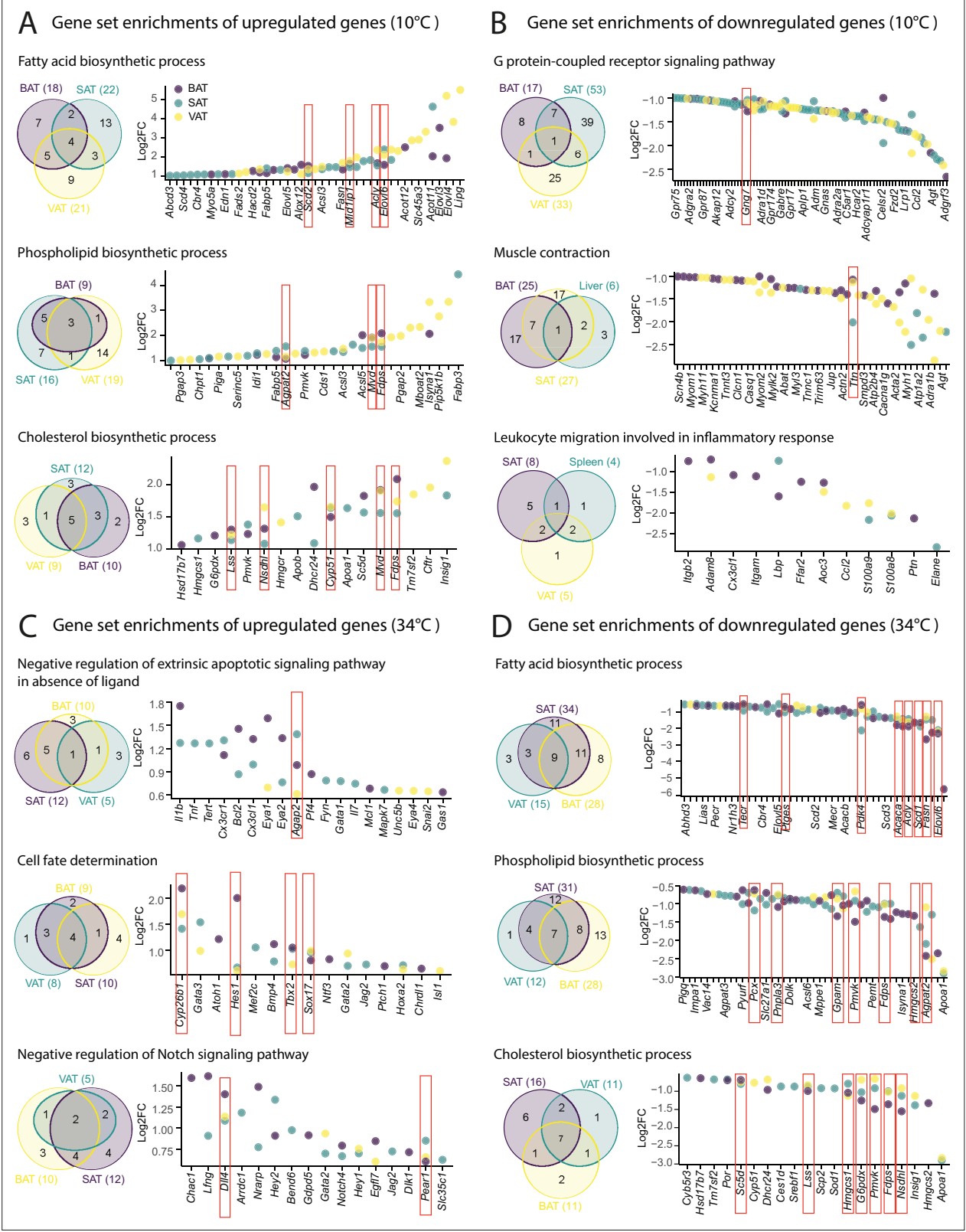

**Figure 7.** Shared tissue GO term enrichments and implicated genes. (**A–D**) Venn diagrams (left) showing shared adipose tissue GO gene set enrichment, and dot plot of genes involved in these gene sets (right) that are significantly (p<0.05) up- (**A–C**), or down-regulated (**B–D**) across adipose tissues from mice as in *Figure 2A*. Not all the genes are labeled on the dot plots. Values are Log2FC and p-values are shown in .

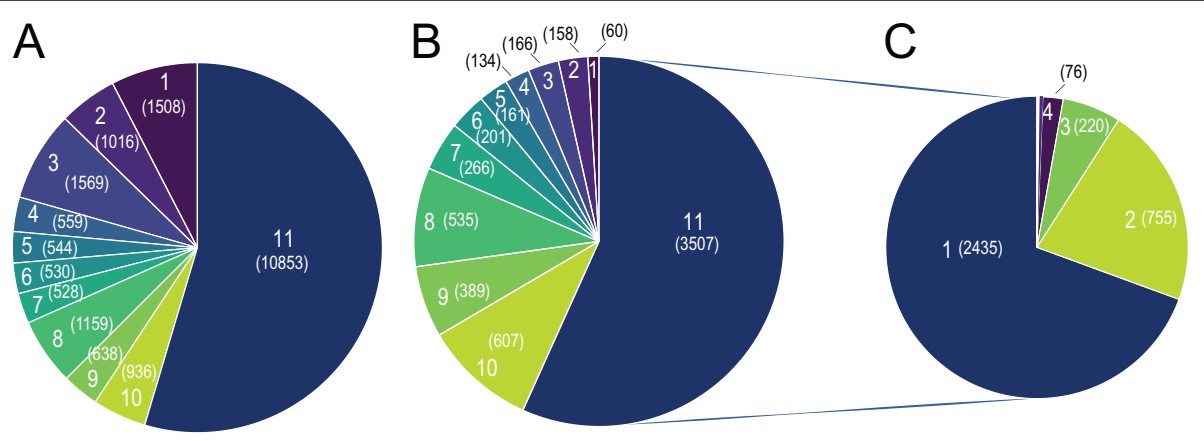

**Figure 8.** Global Tissue-shared and tissue-specific gene expression and regulation patterns across 11 tissues. (**A**) The distribution of 19,881 expressed genes based on the number of tissues. Number of tissues is indicated on each slice of the Pie chart and the number of genes is shown in parenthesis, for example, 10,853 genes are expressed in all 11 tissues and 1,508 genes are expressed in only one tissue (**B**) The distribution of 6183 differentially expressed genes based on the number of tissues. Number of tissues is indicated on each slice of the Pie chart and the number of genes is shown in parenthesis. Number of tissues and colors are kept in the same order as in (**A**), for example, out of the 1508 genes that are expressed only in one tissue, 60 genes are differentially expressed. (**C**) The distribution of 3507 differentially expressed genes based on the number of tissues, for example, 2435 genes are differentially expressed in only one tissue.

acid biomarkers (*Figure 9B*). 139 genes out of the 278 expressed genes were differentially expressed in at least one tissue. We identified 72 tissue-specific gene expression changes, where within ingSAT we found 32 tissue-specific differentially expressed genes ranking it on the top, followed by iBAT with 15 genes and pgVAT with 14 genes. The spinal cord exhibits no tissue-specific differential expression of fatty acid metabolism biomarkers (*Figure 9D*, *Supplementary file 11*). The strong tissue-specificity pattern of differentially expressed genes (*Supplementary file 2*) was further observed in the thermogenic (*Figure 9C*) and the lipid metabolism (*Figure 9D*) biomarkers. Collectively, these results suggest that tissue specificity is dictated in the response to cold (gene regulation level) and not on the gene expression level.

## Tissue-specificity of genes from the same family and with similar functions

By grouping the 19,880 expressed genes of our mouse gene catalog solely based on their gene nomenclature and without considering their specific function, we identified 2827 belonging to a 'group of genes' (*Supplementary file 12*). Members of these groups range from 1084 genes in the biggest group - Olfr (Olfactory receptors), to 2 genes in the smallest group (1273 of these groups have only two members). We found several thermogenic and fatty acid metabolism biomarkers such as Abhd, Abhd, Acsm, Pex, Ppar, Hacd, Elovl, Apoa, etc. as protein families, where we performed comparative analysis of their gene expression and regulation across the 11 tissues (*Supplementary file 12*). Similar as before, we observed tissue-specificity on the response to cold, but not on the gene expression patterns. As an example, we describe the elongation of very long chain fatty acids (ELOVL) protein family, which plays an important role in fatty acid metabolism in adipose tissues. The ELOVL family includes seven genes named *Elovl1* to *Elovl7*, and catalyzes the first and rate-limiting reaction of the very long-chain fatty acid elongation cycle in the fatty acid biosynthesis pathway (*Jump, 2009*). The seven members displayed diverse gene expression levels (*Figure 9A*) and variable rates of differential expression upon cold exposure across the 11 tissues (*Figure 10B*). *Elovl1* and *Elovl7* are present at all 10 tissues with particularly high expression in Ileum, but without any significant change upon cold exposure. The liver expresses all genes except *Elovl7* and *Elovl4* at a very high level, however we found no significant change in the expression levels at cold. The highest average expression value is *Elovl6* in iBAT and the highest regulation was seen for *Elovl3* in ingSAT. *Elovl5* and *Elovl6* expression levels were relatively high across all tissues, and showed a pronounced upregulation by cold in the three adipose tissues (*Figure 10B*). Moreover, we observed a pgVAT-specific increase in *Elovl2* expression upon cold. *Elovl3* on the other hand was upregulated in iBAT and ingSAT, whereas *Elovl4*

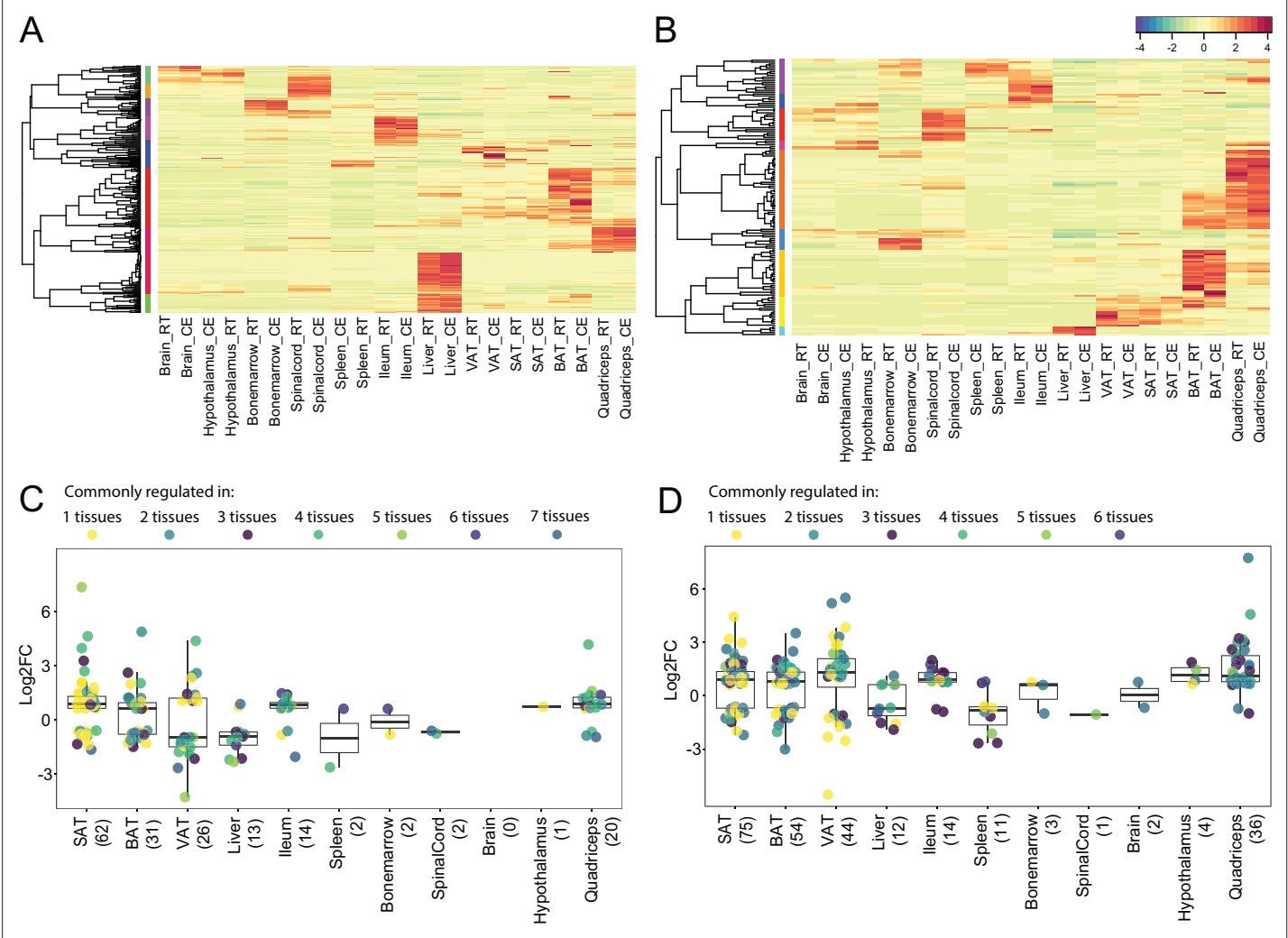

**Figure 9.** Expression and regulation of thermogenic and fatty acid metabolism biomarkers across 11 tissues. (**A**) and (**B**) Heatmap showing an average of the thermogenic and fatty acid metabolism biomarker expression values of mice at RT or cold as in *Figure 2A* across the 11 tissues. (**C**) and (**D**) Log2FC of the significantly (p<0.05) differentially expressed thermogenic and fatty acid metabolism biomarkers across the 11 tissues. Each dot corresponds to the log2FC of a gene and colors specify the number of tissues where that gene is differentially expressed. The total number of differentially expressed genes for each tissue is shown in parenthesis.

expression was enhanced in iBAT and pgVAT from the cold-exposed mice compared to their RT-kept controls and the magnitude of upregulation was much higher in pgVAT (*Figure 10B*). Consistent with our previous analysis, although different members of Elovl gene family were expressed in most of the tissues, their response to cold was tissue specific.

## Discussion

Cold exposure is an extensively studied intervention to promote BAT activity and SAT browning. Despite the potential therapeutic relevance of this environmental trigger in increasing the energy expenditure, the impact of temperature on the transcriptomic landscape of other tissues and their physiological response has received little attention so far. Moreover, different temperatures have been interchangeably used in cold exposure studies. In this study, we first gathered all publicly available RNASeq profiles of C57BL/6J mouse iBATs obtained under different cold exposures ranging from 4°C to 8°C. This comparative analysis revealed that biomarker response to our milder (10 °C) cold setup was effectively in the 50th percentile range with the exception of a few genes. Our results indicate that the variance in magnitude of the transcriptional upregulation across the different studies can be

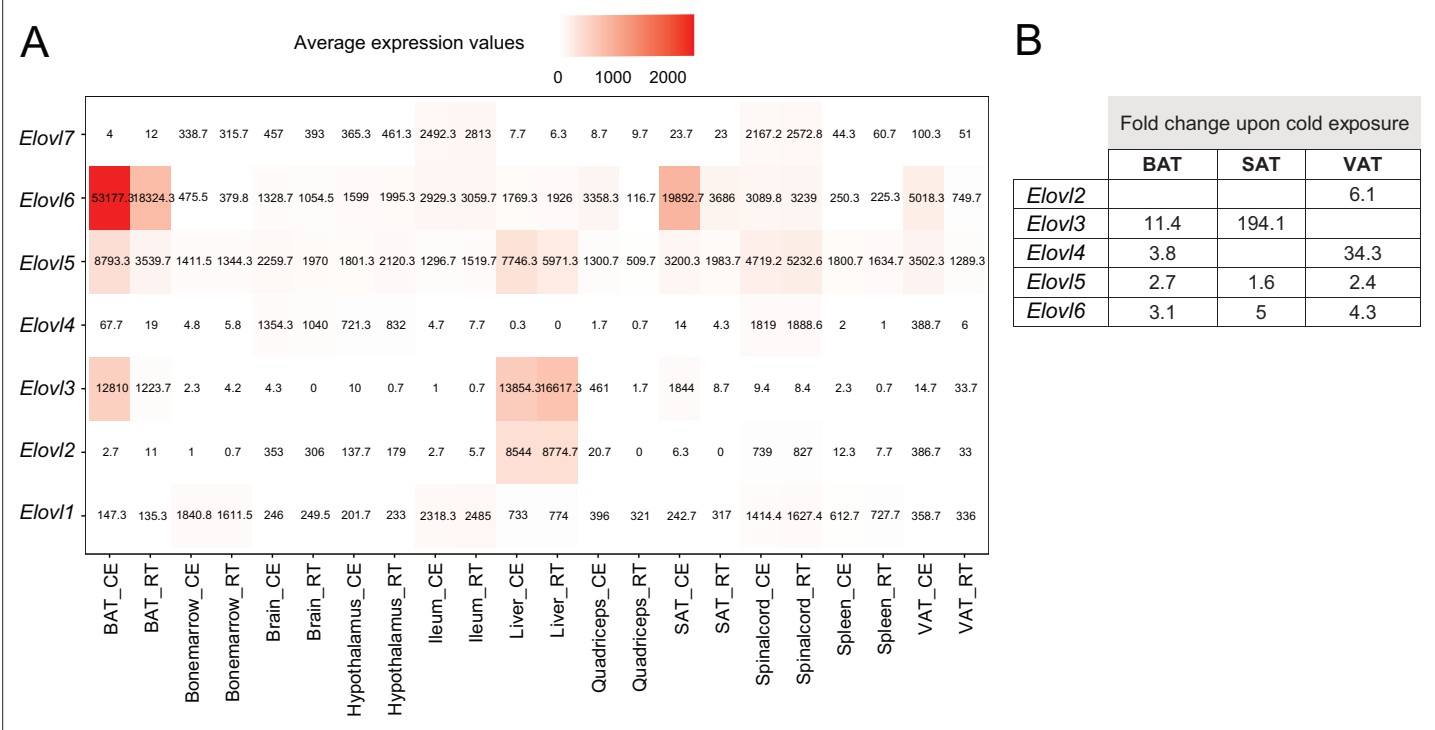

**Figure 10.** Average expression and regulation of Elovl gene family across 11 tissues at RT and cold. (**A**) Heatmap showing average of the normalized gene expression values of mice at RT or cold as in *Figure 2A* across seven tissues where these genes are expressed. (**B**) Log2FC of the significantly upregulated genes (p<0.05) of the Elovl family from mice as in (**A**). Significant changes of the Elovl family were detected only in in the adipose tissues.

explained by the duration of the cold exposure rather than the extent of the temperature decrease per se as evidenced by the expression of the thermogenic and fatty acid metabolism biomarkers. Interestingly, while the regulation of the gene expression was largely proportional to the temperature gradient, we found that temperature close to thermoneutrality (34 °C) caused more pronounced changes than 10 °C, albeit having same 12 °C deviation from the RT (22 °C) in both cases.

When analyzing the comparative RNAseq data across the tissues we observed that as expected, similar tissues effectively clustered together in terms of their overall transcriptomic profile, regardless of the imposed condition: exposure to 10 °C, or 34 °C and compared to 22 °C. The extent of the sample separation following temperature variation was different for different tissues, suggesting variable and tissue-specific responses. These conclusions were further supported by the number and the magnitude of differentially expressed genes and the enriched gene sets. Upregulated GO terms, which were mostly related to thermogenesis, oxidative phosphorylation, fatty acid biosynthesis, elongation and catabolic processes, and mitochondrial ATP synthesis scored the highest in the adipose tissues during cold. This was followed by the hypothalamus and the ileum where some of the upregulated pathways related to lipid metabolism went in the same direction as the adipose tissues, however to a lower extent. Strikingly, the tissue specificity in the response to 10 °C, or 34 °C and compared to 22 °C was not dictated by the gene expression differences between the various tissues, since we found the same levels of alterations even within the commonly expressed genes (55%).

The multi tissue data in this study opens possibilities for integrative analyses of the adaptation to environmental temperatures, and it allows investigating how organs compensate for the increased activity of the above-mentioned biological process. As expected, in the adipose tissues, which during cold showed the highest number of upregulated gene sets particularly related to thermogenesis, we also identified a considerable number of downregulated gene sets. This may suggest that on a tissue level the energy expenditure is balanced by a tendency for lowering a range of biological processes to maximize the necessary biological response to a given trigger. Conversely to 10 °C exposure, the majority of tissues showed higher positive than negative regulation of gene expression and GO terms at 34 °C.

On the other hand, such balance between up- and downregulated biological processes was not seen in the spleen, where we observed primarily downregulated gene sets, which included processes such as innate immune response in mucosa, neutrophil-mediated killing of symbiont cell, inflammatory response, and many other responsive systems, for example response to bacteria. This could be explained by its mainly immunologic functions, a biological process that might be blunted during cold, in line with the concept that maintenance of immune functions is energy costly and thus in competition with other energy demanding programs, including thermogenesis (*Spiljar et al., 2021*). Metabolic and immunologic regulation are tightly interwoven, particularly in the intestine (*Spiljar et al., 2017*), and our data suggests that temperature modulation mainly provokes immunologic changes in the intestine. Intriguingly, our results further indicate that certain pathways are inversely regulated across tissues. For example, the triglyceride catabolic process was upregulated in ingSAT, brain, and hypothalamus, but downregulated in spleen during cold. In part, this may indicate an overall redistribution of the metabolic energy toward the tissues necessary for acute response to an environmental trigger such as cold, on account of the other maintenance biological programs.

Examining tissue-specific and tissue-shared genes and gene set signatures revealed that tissue-specificity both at level of genes and gene sets dominate over tissue-shared patterns. Even within the adipose tissues, we observed preferential tissue-specific response over shared characteristics. Interestingly, the shared-tissue gene sets, for example upregulation of fatty acid biosynthetic process, do not necessarily require upregulation of the same genes between organs. This highlights the tissue specificity also on a gene level, where regulation of different genes contributes to emergence of the same biological process. This is further supported by analysis of the Elovl gene family during cold in context of the different tissues, suggesting that different members of a given gene family evolve and are regulated in a tissue-specific manner. Such inter-tissue specificity of genes from the same family may guide discovery of tissue-specific regulators upon cold adaptation and contribute to a better understanding of the underlying molecular basis of cold adaptation of each tissue. Surprisingly, while cold exposure profoundly altered *Ucp1* expression in the adipose tissues, and also showed increased levels in the quadriceps, which could be due to contaminating or interspersed muscle-resident adipocytes. Follow-up experiments will address the cause of these changes.

Our work provides a full spectrum of thermal adaptation-induced transcriptional reprogramming of multiple mouse tissues, coupled to computational analysis of these changes on a gene and functional level. This resource is made available through a web-based application and database (https://metlabomics.unige.ch/Search), equipped with tools that allow users to navigate, explore, retrieve and compare gene expression regulations across gender, temperature and tissue. Additionally, users can download partial, or complete information as a.csv file to further use for visualization or other intended analyses. In summary, our work shows that 10 °C cold exposure causes representative transcriptomics response in BAT and highlights the temperature-regulated local alternations in the transcriptomic profile across multiple tissues, which include neural, immune, and metabolic responses. We believe that these tissue-specific temperature-induced expression atlas would be a useful resource for studying the physiological alterations in response to lower environmental temperatures in an integrative manner.

## Materials and methods
### Mice

Male 8 weeks old C57BL/6 J mice were obtained from Janvier, France (cold exposed mice and controls) or Charles River, France (warm exposed and control mice). Mice were housed in a specific pathogen-free (SPF) facility in 12 hr day/night cycles with free access to irradiated standard chow diet and water from autoclaved bottles. All mice were housed 2 per cage with a normal use of bedding material, supplemented with aspen wood bars (blocks), but in the absence of nesting material to ensure controlled temperature conditions. Cold exposure was performed in a light- and humidity-controlled climatic chamber (TSE, Germany) under SPF conditions, at 10 °C for 2 weeks with an initial acclimatization period of 5 days at 18 °C and 5 days at 14 °C. Male, 8-week-old C57BL/6J mice were exposed to 34 °C warm (WE) or room temperature (RT) for 4 weeks. All animal experiments were performed at the Universities of Geneva with authorization by the responsible Geneva cantonal, and

Swiss federal authorities in accordance with the Swiss law for animal protection (GE44/20, GE134/18, and GE174/19).

## Experimental model and subject details

Our study design is summarized in *Supplementary file 13*. Transcriptome dataset were generated from 11 different tissues interscapular brown adipose tissue (iBAT), bone marrow, brain, hypothalamus, ileum, liver, quadriceps, inguinal subcutaneous adipose tissue (ingSAT), spinal cord, spleen, and perigonadal visceral adipose tissue (pgVAT) of C57BL/6J mice subjected to cold, or kept at RT. For 10 °C samples, two mice were pooled with a total of six mice per group, except for the spinal cord and bone marrow, where individual mouse tissues of 5 and 6 mice were analyzed, respectively. For 34 °C exposure and the respective room temperature controls, 2 mice were pooled for each organ. The summary of mice and tissue sampling is provided in *Supplementary file 13*.

## RNA extraction for RNA sequencing

After collection, all tissues except bone marrow were snap frozen in liquid nitrogen and stored at –80 °C until used. Frozen tissues were mechanically homogenized with 1 stainless steel bead (5 mm) in 1 ml Trizol (Thermo Fisher Scientific) by shaking for 50 s at 30 Hz (TissueLyser, Qiagen). 200 µL chloroform was added to homogenize Trizol samples, followed by 15 s shaking and centrifugation (15 min, 12000 RCF, 4 °C). The chloroform phase was collected, shaken for 15 s with 500 µL isopropanol and centrifuged (50 min, 12,000 RCF, 4 °C). The pellet was washed with 70% ethanol twice (10 min, 8000 RCF, 4 °C) and dissolved in 50 µL PCR-grade water. Bone marrows were flushed immediately after collection from mouse, cells were spun and loaded onto shredder columns (Qiagen). Shredded bone marrow cells were frozen (–80 °C) in RLT buffer (1% beta-mercaptoethanol) until RNA extraction using RNAeasy mini kit (Qiagen). RNA integrity number (RIN) was determined in all samples (Bioanalyzer 2100, Agilent) before sequencing.

## RNAseq sequencing

The mRNA sequencing was done at the iGE3 Genomics Platform at the CMU of the University of Geneva for bone marrow, spinal cord and quadriceps of the 10°C-exposed mice and their RT controls, as well as all the samples from the 34°C-kept mice and their RT controls, while for all other tissues at the Microarray and Deep-Sequencing Core Facility of the University Medical Center Göttingen. Libraries for sequencing of bone marrow, spinal cord and samples from the 34 °C exposure experiments were prepared with the TruSeq stranded mRNA kit and sequenced with read length SR50 (Illumina HiSeq 4000). For all other tissues RNA-seq libraries were prepared using the NEBNext Ultra RNA Library Prep Kit and were pooled and sequenced on an HiSeq 2000 sequencer (Illumina). The sequencing quality control was done with FastQC v.0.11.5 (http://www.bioinformatics.babraham.ac.uk/projects/fastqc/). The reads were mapped with STAR aligner v.2.6.0c to the UCSC *Mus musculus* mm10 reference. The transcriptome metrics were evaluated with the Picard tools v.1.141 (http://picard.sourceforge.net/). The table of counts with the number of reads mapping to each gene feature of the *Mus musculus* mm10 reference was prepared with HTSeq v0.9.1 (htseq-count, http://www-huber.embl.de/users/anders/HTSeq/).

## Data analysis

Raw counts were processed and analyzed by R/Bioconductor package EdgeR v. 3.4.2, for normalization and differential expression analysis. The counts were normalized according to the library size and filtered. Only genes having log count per million reads (cpm) >0 were kept for the further analysis. After normalization of the counts, transcript abundances were compared in pairwise conditions in a modified Fischer exact test (as implemented in edgeR). Two-tailed unpaired Student's t-test was used for pair-wise comparisons, and p<0.05 was considered statistically significant, unless otherwise specified. Genes were considered significant if they passed a fold change (FC) cutoff of |FC|>1.5 and a p-value ≤0.05, and were further subjected to gene ontology analysis, using R/Bioconductor package topGO (https://bioconductor.org/packages/release/bioc/vignettes/topGO/inst/doc/topGO.pdf), together with Rgraphviz, Pearson correlation similarity analysis, and heatmap visualization. Principal Components Analysis (PCA) and volcano and dot plots were generated in R, and the scripts used for the analysis and generating figures are available at https://github.com/Nhadadi/Mouse_AllTissue_

Transcriptomics, (copy archived at swh:1:rev:ee7742cb4bf5e49ee9d90414c36a06b714a42d69; *Hadadi, 2021*). All generated results are available at https://metlabomics.unige.ch/Search. This online application provides a user-friendly interface to access the data sources, search and filter the results based on several criteria such as significance and fold change, gender, tissue and treatment.

## Acknowledgements

We thank the iGE3 Genomics Platform at the Medical faculty of the University of Geneva, and the NGS- Integrative Genomics Core Unit (NIG) at the Institute of Human Genetics, University Medical Center Göttingen (UMG) for the mRNA sequencing.

## Additional information

### Funding

| Funder | Grant reference number | Author |
| --- | --- | --- |
| Hôpitaux Universitaires de Genève | RC2-09 | Doron Merkler Mirko Trajkovski |
| Schweizerische Multiple Sklerose Gesellschaft | | Doron Merkler Mirko Trajkovski |
| European Research Council | ERC Consolidator grant agreement no. 815962 | Mirko Trajkovski |
| Swiss National Science Foundation | 310030_173010 | Doron Merkler |
| Swiss National Science Foundation | 310030_205042 | Mirko Trajkovski |

The funders had no role in study design, data collection and interpretation, or the decision to submit the work for publication.

### Author contributions

Noushin Hadadi, Conceptualization, Data curation, Formal analysis, Investigation, Methodology, Software, Validation, Visualization, Writing – original draft; Martina Spiljar, Conceptualization, Formal analysis, Investigation, Methodology, Validation, Visualization, Writing – original draft; Karin Steinbach, Melis Çolakoğlu, Claire Chevalier, Formal analysis, Investigation, Methodology; Gabriela Salinas, Investigation, Methodology; Doron Merkler, Conceptualization, Funding acquisition, Methodology, Project administration, Resources, Supervision, Writing - review and editing; Mirko Trajkovski, Conceptualization, Funding acquisition, Investigation, Methodology, Project administration, Resources, Supervision, Writing – original draft, Writing - review and editing

### Author ORCIDs

Noushin Hadadi (ID) http://orcid.org/0000-0001-6614-4910
Mirko Trajkovski (ID) http://orcid.org/0000-0002-3152-9551

### Ethics

All animal experiments were performed at the University of Geneva with authorization by the responsible Geneva cantonal, and Swiss federal authorities in accordance with the Swiss law for animal protection (GE44/20, GE134/18 and GE174/19).

### Decision letter and Author response

Decision letter https://doi.org/10.7554/eLife.78556.sa1
Author response https://doi.org/10.7554/eLife.78556.sa2

## Additional files

### Supplementary files

• Supplementary file 1. 148 potential thermogenic biomarkers combined from literature and GO database, together with their log2FC across 7 studies.

• Supplementary file 2. 292 potential fatty acid metabolism biomarkers from GO database, together with their log2FC across 7 studies.

• Supplementary file 3. Log2FC of significantly (p<0.05) regulated genes across 7 tissues exposed to 34°C.

• Supplementary file 4. Log2FC of significantly (p<0.05) regulated genes across 10 tissues exposed to 10°C.

• Supplementary file 5. Comparisons of the log2FC of common genes across the tissues between the 10°C and 34°C exposures.

• Supplementary file 6. Statistical values for the thermogenic and fatty acid biomarker changes in BAT and SAT after 10°C and 34°C exposures.

• Supplementary file 7. Enriched GO terms across 10 tissues (exposed to 10°C) and 7 tissues (exposed to 34°C).

• Supplementary file 8. 248 commonly enriched GO terms between both treatments (exposure to 10°C and to 34°C).

• Supplementary file 9. Statistical values of the regulated pathways shown in *Figure 7*.

• Supplementary file 10. Average of the normalized gene expression values across the 11 tissues at RT and at 10°C.

• Supplementary file 11. Average of the normalized gene expression values of thermogenic and fatty acid metabolism biomarkers of mice at RT and at 10°C across the tissues.

• Supplementary file 12. Gene family members.

• Supplementary file 13. Summary of the study design.

• Transparent reporting form

### Data availability

The raw counts from the RNA-seq data, and the code for the bioinformatics pipeline developed for this study have been made freely available at https://github.com/Nhadadi/Mouse_AllTissue_Transcriptomics, (copy archived at swh:1:rev:ee7742cb4bf5e49ee9d90414c36a06b714a42d69). The accession number for RNA-seq data reported in this paper is GSE198046.

The following dataset was generated:

| Author(s) | Year | Dataset title | Dataset URL | Database and Identifier |
|---|---|---|---|---|
| Hadadi N, Spiljar M | 2022 | Comparative multi-tissue profiling reveals extensive tissue-specificity in transcriptome reprogramming during thermal adaptation | http://www.ncbi.nlm.nih.gov/geo/query/acc.cgi?acc=GSE198046 | NCBI Gene Expression Omnibus, GSE198046 |

The following previously published datasets were used:

| Author(s) | Year | Dataset title | Dataset URL | Database and Identifier |
|---|---|---|---|---|
| Hao Q, Yadav R, Basse AL, Petersen S, Sonne SB, Rasmussen S, Zhu Q, Lu Z, Wang J, Audouze K, Gupta R, Madsen L, Kristiansen K, Hansen JB | 2014 | Transcriptome profiling of brown adipose tissue during cold exposure | https://www.ncbi.nlm.nih.gov/geo/query/acc.cgi?acc=GSE63031 | NCBI Gene Expression Omnibus, GSE63031 |

*Continued on next page*

*Continued*

| Author(s) | Year | Dataset title | Dataset URL | Database and Identifier |
|---|---|---|---|---|
| Marcher AB, Loft A, Nielsen R, Vihervaara T, Madsen JG, Sysi-Aho M, Ekroos K, Mandrup S | 2015 | The acute cold response of brown adipose tissue analyzed by RNA-seq | https://www.ncbi.nlm.nih.gov/geo/query/acc.cgi?acc=GSE70437 | NCBI Gene Expression Omnibus, GSE70437 |
| Chai X, Yoon MJ, Kim HJ, Kinyui Alice LO, Zhang Z, Xu D, Bai H | 2016 | RNA profiling to describe the transcriptional change during white fat browning, brown fat activity and inactivation | https://www.ncbi.nlm.nih.gov/geo/query/acc.cgi?acc=GSE86338 | NCBI Gene Expression Omnibus, GSE86338 |
| Yiming Cheng LJ, Keipert S, Zhang S, Hauser A, Graf E, Strom T | 2018 | Gene expression signature of brown and inguinal white fat of mice kept at 30°C vs. 5°C | https://www.ncbi.nlm.nih.gov/geo/query/acc.cgi?acc=GSE112582 | NCBI Gene Expression Omnibus, GSE112582 |

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
