## [Editor Report]

This study gene profiled multiple tissues from mice housed at room temperature, cold and mildly warm. The availability of the gene expression datasets should be of interest to the field and may open up new research avenues regarding temperature responses in different tissues.

---

## [Decision Letter]

**Decision letter after peer review:**

[Editors’ note: the authors submitted for reconsideration following the decision after peer review. What follows is the decision letter after the first round of review.]

Thank you for submitting your work entitled "Comparative multi-tissue profiling reveals extensive tissue-specificity in transcriptome reprogramming upon cold exposure" for consideration by *eLife*. Your article has been reviewed by 2 peer reviewers, and the evaluation has been overseen by a Reviewing Editor and a Senior Editor. The reviewers have opted to remain anonymous.

Comments to the Authors:

We are sorry to say that, after consultation with the reviewers, we have decided that your work will not be considered further for publication by *eLife*. Although the work is of interest, the reviewers considered that in the absence of data relating to i) thermoneutrality, ii) muscle, and iii) a Shiny App for interrogating the data in-depth, the findings do not have the significance that we require for publication in *eLife*. However, we encourage you to resubmit to *eLife* as a resource paper if you obtain additional data addressing these 3 concerns. We hope that the reviews copied below help you improve your manuscript.

---

## [Author Response]

[Editors’ note: the authors resubmitted a revised version of the paper for consideration. What follows is the authors’ response to the first round of review.]

Comments to the Authors::We are sorry to say that, after consultation with the reviewers, we have decided that your work will not be considered further for publication by eLife. Although the work is of interest, the reviewers considered that in the absence of data relating to i) thermoneutrality, ii) muscle, and iii) a Shiny App for interrogating the data in-depth, the findings do not have the significance that we require for publication in eLife. However, we encourage you to resubmit to eLife as a resource paper if you obtain additional data addressing these 3 concerns. We hope that the reviews copied below help you improve your manuscript.

We wish to thank you for the careful revision of our manuscript, and for the constructive and critical comments that helped us extend the analysis and improve this resource paper. In the revised version we fully addressed the critical recommendations and included numerous additional data. We are particularly excited with the new data relating to i) thermoneutrality, ii) muscle, and iii) an App that allows in-depth interrogation of the results, as specified below.

i) thermoneutrality

We agree that the typical housing of mice at room temperature (RT) of 22°C, is below their thermoneutrality, and a substantial amount of their energy expenditure is devoted to maintaining core body temperature. The ambient temperatures where metabolic rate is at a minimum is the usual definition of a thermoneutral zone, and it is in the interval of 29°C in the light phase and up to 33°C during dark (Škopet *et al.*, 2020). Therefore, we conducted new systematic transcriptomics analysis in liver, quadriceps, ileum, brain, BAT, SAT and VAT at temperature close to the mouse thermoneutral zone (33-34°C), chosen to match the difference of 12°C between RT (22°C) and cold (10°C). In the revised study we repeated the analysis using both 10°C and 34°C compared to RT, and established how differential expression impacts specific cellular functions across the tissues and identify shared, specific, and inversely regulated gene signatures and gene set enrichments temperature variation. Interestingly, while the regulation of the gene expression was largely proportional to the temperature gradient, we found that temperature close to thermoneutrality (34°C) caused more pronounced changes than 10°C, albeit having same 12°C deviation from the RT (22°C) in both cases.

ii) muscle

We completely agree that muscle was unfairly excluded from the initial analysis, as it is necessary to gain more comprehensive overview on the temperature-driven transcriptomics changes across the tissues. We therefore added quadriceps in both 10°, 34°, and the respective RT controls, and repeated all the analysis with this tissue included.

iii) a Shiny App

Thank you for this excellent suggestion – we agree that it’s the best if a resource manuscript is coupled to an application that allows user to explore the generated data. As there are some issues associated to free version of some apps, such as a limitation of up to 25 active hours per month, number of concurrent users, etc. we decided to take a step further and developed a sustainable web-based application (https://metlabomics.unige.ch/Search) that can be updated with new datasets in the coming years, and that is easier for the users to interact with. The online database is equipped with an interface that allows the users to retrieve all the data reported in the paper by applying several filters on variables such as gender, tissue, temperature gradient, etc. Additionally, users can download partial, or complete information provided on the website as a.csv file to further use for visualization or other intended analyses.